# Brief communication: Organochlorine pesticides in an archived firn core from Law Dome, East Antarctica

Marie Bigot*[1], Mark A. J. Curran[2,3], Andrew D. Moy[2,3], Derek C.G. Muir[4], Darryl W. Hawker[5], Roger Cropp[5], Camilla F. Teixeira[4] and Susan M. Bengtson Nash[1]

[1] Environmental Futures Research Institute, Griffith University, 170 Kessels Rd, Nathan, QLD 4111, Australia
[2] Australian Antarctic Division, 203 Channel Highway, Kingston, TAS 7050, Australia
[3] Antarctic Climate and Ecosystems Cooperative Research Centre, University of Tasmania, Private Bag 80, Hobart TAS 7001, Australia.
[4] Aquatic Contaminants Research Division, Environment Canada, 867 Lakeshore Rd, Burlington ON, L7R 4A6 Canada
[5] School of Environment, Griffith University, 170 Kessels Rd, Nathan, QLD 4111, Australia

*Correspondence to*: Marie Bigot (mbigot1@gmail.com)

**Abstract.** Organochlorine pesticides (OCPs) were, for the first time, quantified in archived firn cores from East Antarctica representative of 1945-1957 and 1958-1967 (current era, C.E.). The core sections were melted under high purity nitrogen atmosphere and the melt water analysed. Methods allowed quantification of hexachlorocyclohexanes, heptachlor, *trans*-chlordane, dieldrin and endrin. While the core presented evidence of nominal contamination by modern-use chemicals, indicating handling and/or storage contamination, legacy OCP concentrations and deposition rates reported are orders of magnitude lower than those from Arctic regions, lending support for their validity. The study further provides a description of equipment used and suggests methods to overcome logistical challenges associated with trace organic contaminant detection in Polar Regions.

## 1 Introduction

Persistent Organic Pollutants (POPs) are ubiquitous, toxic, anthropogenic substances that have been widely used in agriculture and manufacturing industries since the 1930s. A variety of organochlorine pesticides (OCPs) have been listed as POPs under the Stockholm Convention, adopted by the United Nations Environment Program, to ban or severely restrict the use of these chemicals and thereby prevent the environmental burden of these environmental contaminants from becoming greater. By definition, POPs are resistant to environmental degradation processes, allowing them to undergo long-range environmental transport to remote Polar Regions where cold conditions serve to restrict further movement.

Unlike seasonal sea-ice that persists for only limited periods of time, ice sheets are deposited over thousands of years. This makes them a rich archive for the study of past atmospheric composition and climate variations. Levels of inorganic chemical markers such as sodium, potassium, chloride, nitrate or sulphate have previously been successfully determined in glaciology studies (Murozumi et al., 1969). Glacial ice cores also hold temporal information regarding historical organic

pollution events, however their application for this purpose is far less prevalent in the documented literature. Nevertheless, the work by Wang et al. (2008), for example, showed the potential of POPs in this regard by finding a strong correlation between historical usage of POPs in India and POP levels in ice cores of corresponding deposition ages from Mt Everest. To our knowledge, few studies have determined OCP concentrations in glacial ice/firn from the Arctic (Isaksson et al.,
2003;Hermanson et al., 2005;Ruggirello et al., 2010) while none, to date, exist from the Antarctic. Other studies in Polar Regions have focused on other chemicals of environmental interest (i.e. brominated flame retardants, polycyclic aromatic hydrocarbons, current use pesticides and polychlorinated biphenyls) and/or other frozen matrix (i.e. surface snow) (e.g. Fuoco et al., 2012;Gregor et al., 1995;Hermanson et al., 2010;Kang et al., 2012;Zhang et al., 2013;Peters et al., 1995).

In the current global climate context, accelerated melting and subsequent retreat of Antarctic ice shelves are frequently reported. As the ice compartment represents a reservoir of historically deposited POPs, quantification of concentrations in Antarctic continental ice is particularly relevant in order to predict the possible future POP re-emission into the atmosphere and oceans through ice-melt. This secondary input process has previously been demonstrated for melting alpine glaciers in Switzerland (Bogdal et al., 2009). Under this scenario, global efforts to moderate polar ecosystems' exposure to these
potentially hazardous compounds would be compromised.

This work investigates the potential for a well-studied archived Antarctic firn core to elucidate historical deposition rates of OCPs. It was designed as a proof of concept to examine the implications of mismatched sampling priorities between classical glaciology programs and POP research, representing an important step for the assessment of the value of firn core
archives world-wide for the progression of Polar contaminants research. Further, the work addresses some of the challenges associated with the sampling of this matrix for the trace levels of POPs generally observed in Antarctica and describes purpose built equipment for overcoming some of these challenges.

## 2 Materials and methods

### 2.1 Firn core sourcing

For this study, we accessed the historical firn core from a long-established glaciology study site located in East Antarctica: Dome Summit South (DSS), Law Dome (Figure 1). Law Dome is a small independent icecap located in Wilkes Land, East Antarctica and exposed to a maritime climate (Morgan et al., 1997). The DSS site is located near Law Dome summit, approximately 100 km from the coast and at 1370 m elevation (Morgan et al., 1997). This site was selected for its favourable bedrock topography and sufficiently low surface temperatures (mean annual average of - 21.8°C) which preclude summer
melt (Morgan et al., 1997). The DSS site is characterised by a relatively high annual snow accumulation rate of 0.68 m·yr$^{-1}$ water equivalent (weq) (Roberts et al., 2015) facilitating the preservation of very clear seasonal cycles in glaciochemical

species. This provided a means of accurate dating with monthly resolution in the upper portions of the core (van Ommen et al., 2004;Plummer et al., 2012;Roberts et al., 2015).

The 1196 m DSS ice core was collected in 1988-1993 and the entire records spans approximately 80 kyr B.P. (van Ommen et al., 2004). Its chronology is derived from a combination of direct layer counting (van Ommen et al., 2004;Plummer et al., 2012;Roberts et al., 2015) and age ties to other records (van Ommen et al., 2004). This ice core has provided a wealth of atmospheric circulation, ambient temperature, snow accumulation, and climate proxy records for climate reconstructions (Souney et al., 2002;Plummer et al., 2012;Roberts et al., 2015). The sections of the core used in this work were extracted using thermal drilling (Morgan et al., 1997).  Following collection, the 18 cm diameter core was cut to one meter lengths on site, stored in double plastic bags at -18˚C, and archived in large cold storage facilities in Tasmania, Australia, until analysis.

There is an inherent incompatibility between standard glaciology and POP sampling procedures, particularly in terms of the volumes of matrix required for robust analysis, the type of materials used during sampling and long term storage requirements. Nevertheless, given the massive logistical investment to sample ice in remote Polar environments, and the rich source of ancillary information accompanying the well-studied DSS core, access to the remaining firn core represented an opportunity to examine the implications of method incompatibilities and assess whether useful information could be derived from this, and therefore similar valuable archives, world-wide.

Sample volumes recently used for determination of OCPs in other Antarctic frozen matrices such as continental surface snow (Kang et al., 2012) and sea-ice (Dickhut et al., 2005) were 500 mL and 95.5 to 132 L, respectively. In the Arctic, Hermanson et al. (2005) and Ruggirello et al. (2010) used a minimum of 11 L of melted ice/firn core, while Isaksson et al. (2003) used a total of 48 metres of a core (of unknown diameter) separated into nine individual samples. No other previous study on OCPs has been published for continental firn cores in Antarctica, hence the chemical burden and consequent volume of firn required for confident detection was largely unknown. We therefore adopted a conservative approach by utilising just two large volume samples from a known period as opposed to a higher number of samples of smaller volume. The DSS firn core was selected as it yielded large volumes for the relevant deposition time periods of interest, enabling guidance on chemical deposition rates for future investigations. It was separated into batches of approximately 10 metre length each, representing two early OCP deposition periods: 1945-1957 current era (C.E.) (35.4 to 45.6 m depth, sample A) and 1958-1967 C.E. (26.3 to 35.4 m depth, sample B). Density of these firn sections had previously been estimated to be between 0.63 to 0.74 g·cm$^{-3}$ (for core depth between 21.5 to 47.6 m) (Morgan et al., 1997) and dating had been conducted by direct layer counting.

## 2.2 Sampling procedures

Firn core samples were minimally exposed to laboratory ambient air, and whole portions were melted together in a clean stainless steel unit purpose designed and built for POP analysis (Figure 2, 125 cm (length) x 80 cm (height) x 40 cm (depth), approximately 70 kg empty). Melted sample volumes were 144 and 133 L for samples A and B, respectively. These were obtained through melting of two successive batches for each due to the limited capacity of the melting unit. This unit, derived from a design by Gustafsson et al. (2005), allowed complete isolation of the sample and melting under high purity nitrogen atmosphere (Figure 2). The unit was fitted with a drain connected to a modified Kiel in-situ Pump (KISP, Aimes GmbH, Germany), previously used for the purpose of trace organics sampling in water (Petrick et al., 1996). This set-up was designed to provide a means of *in-situ* sample extraction for field based snow, ice and water campaigns. The ice-melting unit is fully transportable for direct sample loading and sealing in the field. The unit can then be brought to the operational Polar research station for access to nitrogen gas, water and electrical power to complete sample extraction. Alternatively, it could be powered on site using generators for in-situ melting.

Melt-water was pumped directly from the melting unit through a pre-furnaced (12h at 450°C) glass fiber filter (GFF, 142 mm, 0.7 µm, Whatman, England) collecting suspended particles followed by a stainless steel cartridge filled with approximately 150 mL of pre-cleaned Amberlite XAD-2 resin (Supelpak$^{TM}$-2, Sigma-Aldrich Co. LLC.) collecting the dissolved fractions of the compounds. The flow rate of melt-water through the system was set at 150 mL·min$^{-1}$. Full sampling material preparation methods and analytical procedures for filters and XAD have been described elsewhere (Bigot et al., 2016). Sample extracts and blanks were analysed by Australian Laboratory Services (ALS Global, Burlington, ON, Canada) for OCPs using gas chromatography high resolution mass spectrometry (GC-HRMS) following standard protocols based on EPA method 1699.

## 2.3 Quality Assurance / Quality Control

The samples were handled in a clean room (positively pressured, high efficiency particulate and carbon filtered air) during extraction at Environment Canada (Canadian Centre for Inland Waters (CCIW), Burlington, ON, Canada). Three blanks (XAD and filters) were treated in the same manner as the samples, i.e. exposed to ambient air during changing of sampling materials in the laboratory, transported together, and extracted and analysed in parallel with samples. All sample values reported are blank corrected using the average of three blanks, providing a means of correcting for possible contamination during laboratory, storage and/or transport of the samples.

True blanks (i.e. similar volume of archive samples representative of a deposition period pre-OCP production) were not available. In order to evaluate possible contamination, modern usage POPs, namely polybrominated diphenyl ethers (PBDEs), were used as markers of contamination resulting from the sampling, processing and/or storage of firn cores.

PBDEs are commercial flame retarding compounds widely produced since the 1970's. Their concentrations were determined in the final extracts using low resolution GC-MS at CCIW/Environment Canada.

Method detection limits (MDLs) were calculated for each compound as three times the standard deviation of the laboratory blanks. Recovery rates were evaluated via spiking sample materials with 200 µL of a solution containing δ-HCH and $^{13}C_8$ Mirex (10 pg·µL$^{-1}$ each), and BDE-71 (9.6 pg·µL$^{-1}$). XAD cartridges were spiked prior to sampling and filters were spiked prior to extraction. Labelled target analytes were also added by ALS Global and responses were used to correct for losses during clean-up and instrument drift. A list of recoveries for all surrogate standards is available in the supplementary information. Results originally expressed in pg·L$^{-1}$ are presented as mean deposition per year (pg·cm$^{-2}$yr$^{-1}$) calculated using the surface area of the core as the deposition surface, thus allowing comparison of the two time periods studied.

## 3 Results and discussion

The relatively large volumes of firn analysed in this study enabled the quantification of the legacy OCPs α- and γ-HCH, heptachlor, trans-chlordane, dieldrin, and endrin in the dissolved fraction of the melt water of both samples (Figure 3). These compounds are likely to be deposited on snow after long range atmospheric transport from source regions within the Southern Hemisphere. Their presence have recently been reported in seawater and/or air close to the coast of East Antarctica (Bigot et al., 2016). Both samples, however, also contained traces of our storage and handling contamination markers, PBDEs, indicating that sampling and/or storage conditions of the archived firn cores introduced organic contamination. These contaminants could have been introduced through various means such as the operator's personal clothing, storage in plastic bags, and the use of contaminated coring tools and inappropriate cleaning methods.

While no direct link can be established between PBDE levels introduced and possible contamination of the firn core by legacy OCPs, the presence of the PBDE markers necessitate caution in the interpretation of results. Notwithstanding this, the firn deposition rates were below 0.8 pg·cm$^{-2}$·yr$^{-1}$ for α- and γ-HCH, heptachlor, trans-chlordane and endrin, while dieldrin had deposition rates of up to 4 pg·cm$^{-2}$·yr$^{-1}$. Previous studies reported data in melt-water concentrations rather than deposition rates. The OCP deposition rates that we report here are derived from melt-water concentrations of < 60 and 310 pg·L$^{-1}$, respectively (for detailed results see Supplementary Information, Table 2). Although the DSS site is characterised by higher snow deposition rates (0.68 m·yr$^{-1}$ weq) compared to the three other Arctic sites for which OCPs in glacial ice/firn cores were documented (from 0.36 to 0.52 m·yr$^{-1}$ weq), OCP results from the DSS site are 10 to 1000 fold lower than the reported levels in the Arctic (Hermanson et al., 2005;Isaksson et al., 2003;Ruggirello et al., 2010). This is consistent with the uneven distribution of the world's past usage of OCPs (Voldner and Li, 1995), with the Northern Hemisphere having contributed larger emissions than the Southern Hemisphere, suggesting a larger pool of OCPs would have reached the Arctic in comparison to Antarctica. Data previously reported from other environmental matrices, i.e. seawater and air, also show a 10

to 100 fold decrease between Arctic and Antarctic concentrations of HCHs, thus lending further support for the validity of the OCP results obtained.

The relatively high sample volumes required for this analysis limit the temporal resolution of results and therefore the detail
with which we are able to investigate accumulation history of OCPs. The samples represent firn from Law Dome over consecutive 13 and 10-year periods, respectively. Observed differences between the two time periods may reflect different usage patterns but may also be a function of the physicochemical properties of individual pesticides. Given OCPs are mainly transported to Polar Regions via atmospheric transport (Wania and Mackay, 1996), the Law Dome area would have been impacted by air originating from the Southern Hemisphere. Access to detailed historical usage of OCPs in this part of the
world is limited, however most legacy OCPs were first used in the 1940's or early 1950's, e.g. in Australia and Africa (Harrison, 1997;Wandiga, 2001). The concentrations reported represent deposition of OCPs when their usage was at or near their maximum, before the first controls on the uses of these compounds were implemented in the 1970's. As a result, it gives an insight on possible maximum concentrations in Antarctic firn that may remobilise from melt water. Dieldrin and $\gamma$-HCH show a distinct increase in their concentrations in sample B, spanning the period (1958 to 1967 C.E.), compared to
those in Sample A (1957 – 1945 C.E.) (Figure 3). This could reflect an increase in Southern Hemisphere usage of these two compounds between 1945 and 1967 C.E. Alternatively, the compounds may have started to deposit later than 1945, in which case sample A's results may be affected by dilution of the deposited burden by uncontaminated sections of the core.

HCHs are the most commonly studied OCPs in the abiotic Antarctic marine environment. Concentrations reported in this
study are 19 and 22 pg·L$^{-1}$ for $\alpha$-HCH  and 22 and 60 pg·L$^{-1}$ for $\gamma$-HCH, which are in the lower range of findings in surface snow samples collected more recently in Antarctica (Kang et al., 2012). The relative contributions of each isomer ($\gamma > \alpha$) observed in the continental firn samples from Law Dome are also consistent with reported Antarctic sea-ice and atmospheric profiles (Dickhut et al., 2005). All OCPs detected in this study have also been recently reported in atmospheric and oceanic samples in the same sector of Antarctica (Bigot et al., 2016). Some of these OCPs, heptachlor and dieldrin in particular, are
currently found in Australian air samples at concentrations amongst the highest reported globally (Wang et al., 2015).

Dieldrin was detected at the highest levels in both dissolved and particulate fractions of the melted firn samples analysed in the present study. It is a versatile cyclodiene insecticide that has been largely used around the world for treatment of soil for plantations, as well as for control of various pests. Its usage was largely restricted from the 1970s, but its use was only fully
banned internationally in 2004, under the Stockholm Convention. In the dissolved fraction, we report an average concentration of dieldrin of 222 and 310 pg·L$^{-1}$ in sample A and B, respectively. The dissolved fraction accumulated in firn cores may, if released due to rapid continental ice melting, be available for uptake by marine organisms, and present a toxicological threat to the Antarctic food web. Dieldrin is currently one of the most prevalent OCPs found in this region (Bigot et al., 2016). Our findings indicate that ice-melt is likely to result in particularly high dieldrin enrichment compared to

other OCPs found in the firn samples. Interestingly, it is also the only compound detected on particles contained in melt-water of the archived ice core at 3.8 and 7.1 $pg·L^{-1}$ for sample A and B, respectively, which suggests it may also be transported on atmospheric aerosols that could originate from wind eroded soils for example. Dieldrin was the only OCP compound detected on aerosols sampled in adjacent areas of the Southern Ocean (Bigot et al., 2016).

This proof of concept study determined OCP concentrations that reached Antarctica during their early usage. Melted water volumes of 144 and 133 L for each sample allowed quantification of legacy OCPs that have previously been reported in the vicinity of Antarctica, however results should be considered as maximum possible values given possible contamination introduced during sampling and/or storage. Results for other OCPs were below MDLs, although higher volumes may

facilitate quantification. HCHs and dieldrin were found at the largest concentrations suggesting that they could be targeted in future Antarctic glacial ice investigations to obtain more refined measurements using much lower volumes.

## 4 Conclusion

Results show that legacy OCPs could have accumulated at Law Dome in deep firn dated from as early as 70 years ago. Our current understanding of organic contaminant retention, mobility and fate during aging of snow and formation of ice is poor.

Apparent concentrations in glacial ice may not accurately indicate historical deposition. The analysis of ice or firn cores may nevertheless give us valuable information on the current OCP reservoir in Polar Regions. Although East Antarctica currently exhibits a slight positive ice mass balance as opposed to the rest of the continent (Sasgen et al., 2013), driven by climate conditions in the Pacific (Meehl et al., 2016), the potential for OCPs to remobilise into the atmosphere and oceans raises the need for development of POP research in the cryosphere. While we found evidence of nominal PBDE contamination

throughout the core, OCP concentrations and deposition rates reported are orders of magnitude lower than those from sites with lower annual snow accumulation in the Arctic. This appears realistic and therefore suggests that our results can be taken as first indications of OCP contamination at Law Dome, East Antarctica, through this time period.

Standard glaciology sampling procedures for inorganic analytes (e.g. metals), climate tracers and those for POPs are not

necessarily compatible, particularly in terms of the volumes required for analysis, and methods used for collection and storage. Given the potential value of POP quantification in the Polar cryosphere, we advocate pro-active sampling in collaboration with glaciology programs to ensure sufficient volumes of dedicated samples according to POP work QA/QC procedures. In particular, plastics should be avoided at all times as a general QA/QC measure for sampling, storage and analysis of such compounds. The ice-melting unit used for the present work is designed to be transportable and represents a

comprehensive tool offering a means of storage, transport, melting and pumping of large volumes of ice. It provides a cost-effective solution to the logistical challenges of transporting these volumes of ice back from Antarctica. It also reduces risks for contamination during storage and transport as well as minimising overall handling of the samples. The melting process

can be performed in-situ provided appropriate power sources are available, or locally at the closest research station. A similar in-situ melting technique has previously been successful for the collection of sea-ice in the Arctic (Gustafsson et al., 2005). These methods should be implemented when dealing with trace POP analysis in Polar Regions. In lieu of such support, however, glaciology archives may provide useful indications for remote locations from which information is still

lacking.

*Acknowledgements*. This study was funded by Australian Research Council Discovery project DP140100018 and the Australian Antarctic Division (AAD) provided funding and the DSS firn core samples (AAS project # 4061). Marie Bigot acknowledges a PhD scholarship from Griffith University. The authors would like to thank the field teams that collected

samples in 1988 - 1993, as well as Meredith Nation, Lauren Wise, Debbie Lang and all personnel that provided assistance at the Australian Antarctic Division. We also thank Whitney Davis and Ron MacLeod at ALS Global for support provided.

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

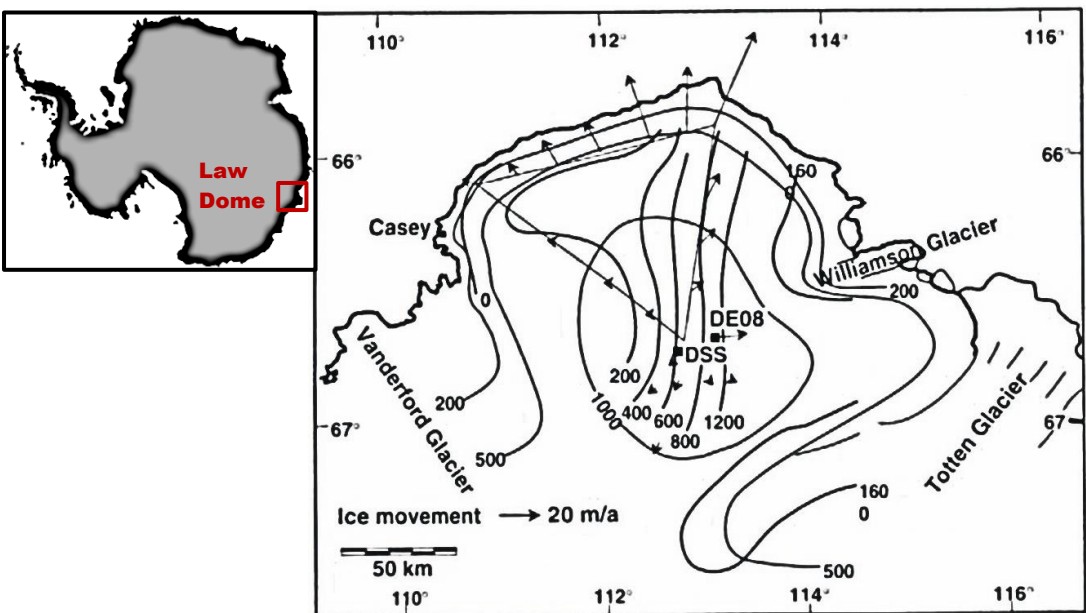

**Figure 1: Location of the Dome Summit South (DSS) coring site at Law Dome in East Antarctica. Elevation contours are represented using thinner lines compared to accumulation contours (expressed in kg·m⁻²) (right hand side map courtesy of Morgan et al., 1997, adapted from Xie et al., 1989).**

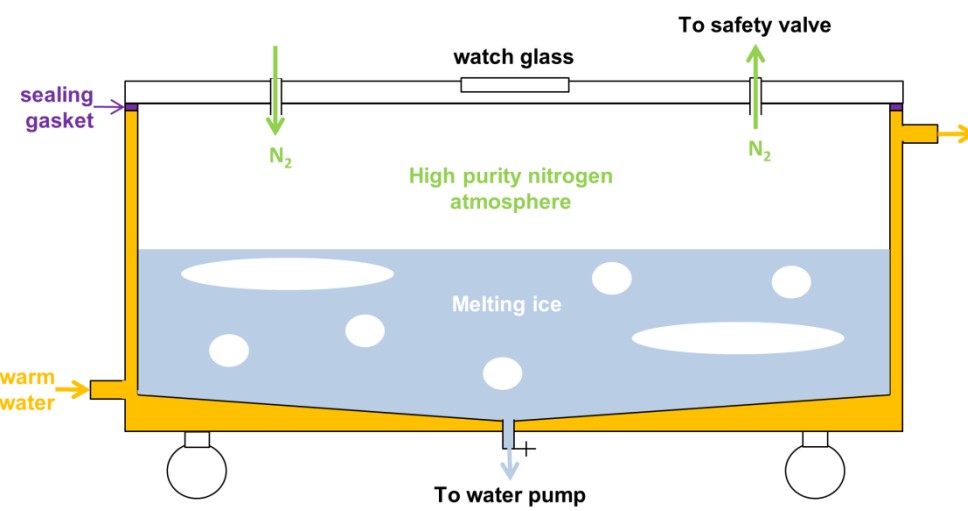

**Figure 2: Conceptual cross sectional diagram of the ice-melting unit used in this work**

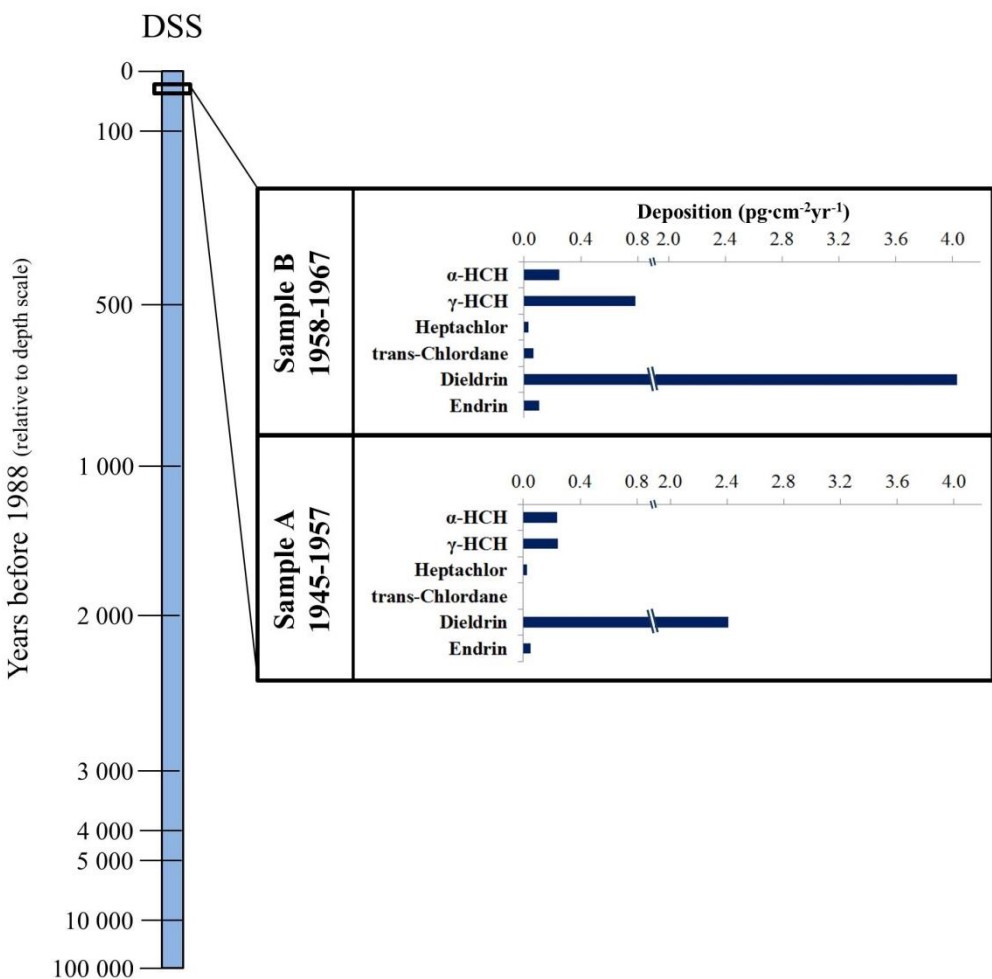

**Figure 3: Deposition rates of 6 legacy OCPs at the Law Dome site in Antarctica between 1945 and 1957 (Sample A) and 1958 to 1967 (Sample B). Note: results should be considered as maximum possible values.**

