# Peer review of "Brief communication: Organochlorine pesticides in an archived firn core from Law Dome, East Antarctica"

_The Cryosphere, 2016_

## Referee Comment (RC1) · Anonymous Referee #1 · 8 Aug 2016

The paper describes the methods used and the results from analyzing two bulk samples from the upper sections from an ice core from Law Dome for Organochlorine pesticides (OCPs). There has been no similar studies from Antarctica published before so therefore this is an important contribution to our knowledge about the fate of contaminants in the atmosphere. The paper is well written and its main focus in on the analytical methods; basically how to handle ice core samples that has been archived for a long time. Naturally, this is important knowledge as such studies described here will likely be on similar situations; i.e. left-over material after some of the fundamental sampling and analyses for dating and climate studies have been completed.

However, before the paper is accepted I would like the authors to provide more and accurate glaciological information about the ice coring site which is completely lacking and are of importance for any studies using material from ice cores.

[Figure]

- The position of Law Dome is likely not familiar to many readers and it is important to include information about where it is situated, altitude, annual accumulation rate and mean annual temperature. All these factors are likely to have an impact of the fate of contaminants during transport, deposition and post-deposition and thus on the concentration found.

- From glaciological perspective Law Dome is not part of the "East Antarctic Ice Shelf"; an expression that is used in this paper which is not correct. There are many ice shelves around the Antarctic continent but they are separate from each other and thus do not form a unit, which is implied here. Also, despite its coastal location Law Dome is not part of any ice shelf. It is a local ice dome with an altitude of close to 1400 m which is much higher that any ice shelf and therefore it is very confusing when melting ice shelves are referred to both in the "Introduction" and "Conclusions". Furthermore, it is not unlikely that the sources and transport paths to Law Dome and lower elevation ice shelves are different and thus it is not a good idea to mix these things freely in the discussion.

p. 5, line 3-4. "Both samples, however, also contained traces of our storage and handling contamination markers, PBDEs, indicating that sampling and/or storage conditions of the archived firn cores introduced organic contamination." I would like to see some more information/ideas around this aspect. For instance, is the type of plastic bags used important?

Finally, "polar regions" should not be capitalized as now is done through the manuscript.

I recommend that the paper is accepted after these comments are considered.

---

## Referee Comment (RC2) · Anonymous Referee #2 · 23 Aug 2016

Review of Bigot et al., "Organochlorine pesticides in an archived firn core from Law Dome, East Antarctica" from "The Cryosphere Discussions", doi:10.5194/tc-2016-178.

This paper documents the results of analysis of organochlorine pesticides in a firn/ice core from Law Dome, Antarctica. The core was drilled several years before analysis, and was kept in storage. There are few studies documenting the accumulation of organic contaminants in ice (of any kind) from Antarctica that were transported by the atmosphere. The reason for so few studies are many, including the lack of available cores, which itself is related to the difficulty with drilling, retrieving and storing cores from Antarctica. Another issue is the expense involved with analytical work. The work of the authors in this regard is greatly appreciated.

This paper should be given consideration for publication after addressing the general

and specific comments made below.

General comments:

In my opinion, the paper addresses a relevant issue within the scope of TCD. The idea presented and tools used in the research are not novel, but the data are new and are a contribution to understanding levels of contamination found in the Law Dome area of Antarctica.

The conclusions are generally sound, although rewriting should be considered. Yes, the results are from as early as 70 years ago, but that results from inherent poor dating resolution of organic contaminants in ice cores. The quantitative difference between OCP amounts found the Antarctic and the Arctic are important, but the authors need to be much more precise in P6L30+ where they state that "deposition rates are orders of magnitude lower than those from Arctic regions". In that statement, they are suggesting that the Arctic is one uniform region with regard to OCP deposition rates, and the published literature shows that this is not true (some of the publications showing this are not included in the reference list). Within the conclusions, I disagree with the use of the word "in-situ" with respect to melting techniques being supported by the references used. I am not certain that the referenced investigations used "in-situ" melting the way it was done for the current investigation because not all of the references specifically state melting at a field site.

The scientific methods and assumptions are clearly stated, along with some of the limitations. I am a bit concerned that limitations involved with using the KISP have not been identified, including long running times and short battery life. However, these are only a problem when using the KISP in the field, instead of in a laboratory.

The results have not been over-interpreted, which is important. The results here are very limited, but are still significant.

It would likely not be possible to reproduce these results, which is a common issue with
environmental work on this level. There is very little related work in Antarctica, and it appears that the authors have given credit to previous work.

I have one question regarding the title: Is the DSS core really firn to a depth of 45.6 meters? There is no mention of the transition depth from firn to ice in this paper.

In the abstract, I do not understand what is meant by "nominal modern-use chemical contamination". How do deposition rates "orders of magnitude lower than those from Arctic regions" support validity? This is not standard QC procedure.

In general, the paper is well organized and carefully written, with the issues noted elsewhere. I note that the use of C. E. in the paper (including the abstract) is not define, requiring that the reader understand what this means. That may not be the case.

Figures: Figure 1 needs additional information, including the dimensions of the unit. It would be useful to know the grade of N2 used and its circulation within the device, and not just that it is a "clean atmosphere".

There is a significant issue with references used in the paper. Why is the Legrand et al. 1984 reference used for aerosols in a glacier? Why not Murozumi et al., 1969 (GCA, 1969, 33, 1247-)? Murozomi et al. also had data for contaminant lead both from Greenland and Antarctica, and was the first paper to identify contamination of ice cores by long-range transport of an anthropogenic substance. This issue takes on greater relevance with the earlier studies about organic contaminants in the Arctic used as references in this manuscript. One assumes that the Gregor et al. (1995) paper is used as a reference because it was the earliest study on PCB deposition in an Arctic glacier. This assumption arises because of absence of later reference to PCBs in glaciers. So again, why use Legrand et al., 1984, as a reference if it is not the earliest?

There is no reason (or reference) given for the quantification of PBDEs as evidence of contamination. What is the rationale behind this?

Specific comments & technical issues:

P1L26: Replace "hereby" with "thereby". The statement made here, to "minimize the environmental and human health hazards that they pose" is overstated. Even though the compounds on the Stockholm list have been banned or restricted, they are still found in the environment. And they are still moving around. The only thing Stockholm can accomplish is prevent the mass of these contaminants now in the environment from becoming greater.

P1L33: Again, Legrand et al., 1984 is not the best reference in this context.

P2L5: The claim that "only one study has documented OCP concentrations in glacial ice/firn from the Arctic" is not correct. I can immediately think of 3 without looking.

P2L6: To say that there are no OCP studies in firn/ice cores from Antarctica is splitting hairs a bit too much for work like this. What about Kang et al., 2012, from the reference list?

P5L9: Again, this limited selection of references is a bit surprising considering those not mentioned. Why are no comparisons offered between the results of Kang et al. 2012 and the results of this manuscript?

P3L4: This repeats P2L2.

P3L13: Apparently it is true that Isaksson et al. (2003) never mention the diameter of the core used. The current manuscript also never mentions the diameter of the Law Dome core.

P3L15: While it may be true that no earlier firn core studies are available to use as a guide for sample volume needed from Antarctica, the authors could have used Kang et al. study on surface snow as a guide.

P3L19: C. E.?

P3L26: Was the system shown in Figure 1 capable of holding 144 L of melt?

P4L27: The original results were not "corrected to estimate the mean deposition". It is not a matter of correction, but calculation.

P4L28: In my dictionary, "basal area" is defined as the area of total tree trunks (diamters) as a fraction of given land area where the trees are growing. That does not seem to apply here.

P4L31: What is meant by "dissolved fraction of the melt water"?

P6L7: Reference to Stockholm for ban on Dieldrin is not very good. Dieldrin was banned under other regulations many years before 2004.

P7L8: The authors need to do a better job describing "in situ".
* * *

---

## Referee Comment (RC3) · Anonymous Referee #3 · 25 Aug 2016

In order to understand better the magnitude and to limit the environment's damage, it is clearly necessary to know about transportation, accumulation and concentration changes of organic pollutants through ages. One irreplaceable source to obtain some of this information is the investigation of chemical compounds in polar ice cores. Since snowflakes have the capacity to adsorb gas-phase chemical compounds, ice-sheets represent an archive of notions of the varying deposition of trace chemicals. From the stratigraphy of the ice cores, the ice-sheets can be temporally related, and every core can cover the chemical history of many decades. Antarctica's low temperatures during the whole year both reduce the speed of chemical reactions and allow the ice layering mentioned above. The paper, well written, describes the methods and the results from two bulk samples from the upper sections from an "old" ice core from Law Dome for organochlorine pesticides: few studies have been published, so the paper is another

small step towards knowledge about contaminants transport mechanism in the atmosphere. The ice core used in this study was drilled on Law Dome, a small ice cap with independent ice flow located on the edge of the main East Antarctic ice sheet (a map could be useful for many readers). The characteristics of Dome Summit South (DSS), include a high annual accumulation rate (0.7 m/yr ice equivalent), relatively low mean surface temperatures (-21.8 °C), and low wind speeds (8.3 m/s). These site characteristics lead to highly resolved records with clear annual cycles in most measured parameters, giving very robust chronological control. I did not find any of this information on the paper, and I have to find myself, in the literature, despite their fundamental importance. As RC1 wrote, it is also absolutely important to have more information around storage/handling contamination, but not only, the sampling conditions must absolutely be considered and described: was it a manual drilling or not? I think not, the ice-drilling was too deep (1196 m), this "modus operandi" can have influenced the results. The relationship between Arctic and Antarctic pollutants concentration amount may be taken into consideration, but cannot become the key to any conclusions. While it may be true that no earlier firn core studies are available to use as a guide for sample volume needed from Antarctica for OCPs, I think the volume used is really too big, a tenth could be enough for a more defined (in time) measurement.

P3L19: C. E.?

I recommend that the paper is accepted after these comments are considered.

---

## Author Comment (AC1) · 18 Sep 2016

Manuscript number: tc-2016-178 Manuscript type: Brief Communication Title: "Brief communication: Organochlorine pesticides in an archived firn core from Law Dome, East Antarctica"

Response to comments from Reviewer 1

The authors would like to thank the anonymous reviewer for their very constructive comments.

- Site-specific information has been added from p2, line 26 to p3, line 2. "Law Dome is a small independent icecap located in Wilkes Land, East Antarctica and exposed to a maritime climate (Morgan et al., 1997). The DSS site is located near Law Dome

summit, approximately 100 km from the coast and at 1370 m elevation (Morgan et al., 1997). This site was selected for its favourable bedrock topography and sufficiently low surface temperatures (mean annual average of - 21.8°C) which preclude summer melt (Morgan et al., 1997). The DSS site is characterised by a relatively high annual snow accumulation rate of 0.68 metres (water equivalent) (Roberts et al., 2015) facilitating the preservation of very clear seasonal cycles in glaciochemical species. This provided a means of accurate dating with monthly resolution in the upper portions of the core (van Ommen et al., 2004;Plummer et al., 2012;Roberts et al., 2015)."

- Thank you for the clarification regarding Law Dome not being part of the "East Antarctic Ice Shelf". All references to the East Antarctic Ice Shelf have been removed from the manuscript.

- Reviewer wrote: "p.5, line 3-4. "Both samples, however, also contained traces of our storage and handling contamination markers, PBDEs, indicating that sampling and/or storage conditions of the archived firn cores introduced organic contamination." I would like to see some more information/ideas around this aspect. For instance, is the type of plastic bags used important?" We have further expanded on this point firstly on p5, lines 14-16 "These contaminants could have been introduced through various means such as the operator's personal clothing, storage in plastic bags, and the use of contaminated coring tools and inappropriate cleaning methods." Then on p7, line 28-29 "In particular, plastics should be avoided at all times as a general QA/QC measure for sampling and analysis of such compounds."

- Reviewer wrote: " "polar regions" should not be capitalized as now is done through the manuscript." In the case of this manuscript, we purposely used capital letters for "Polar Regions" to distinguish "polar" from its other meanings in chemistry.

Please find latest manuscript draft in attachment.

Please also note the supplement to this comment:

http://www.the-cryosphere-discuss.net/tc-2016-178/tc-2016-178-AC1-supplement.pdf

[Figure]

**Supplement:**

**Brief communication: Organochlorine pesticides in an archived firn core from Law Dome, East Antarctica**

Marie Bigot\*1, Mark A.J. Curran2,3, Andrew D. Moy2,3, Derek C.G. Muir4, Darryl W. Hawker5, Roger 5 Cropp5, Camilla F. Teixeira4 and Susan Bengtson Nash1

[revised manuscript text omitted]

---

## Author Comment (AC3) · 18 Sep 2016

Manuscript number: tc-2016-178 Manuscript type: Brief Communication Title: "Brief communication: Organochlorine pesticides in an archived firn core from Law Dome, East Antarctica"

Response to comments from Reviewer 3

The authors would like to thank the anonymous reviewer for their thoughtful comments and suggestions.

- Thank you for your suggestions regarding information on site-specific characteristics. They have been added to the revised manuscript as previously addressed in our response to Reviewer 1.

[Figure]

- We have added a map (see new Figure 1) which gives a better idea of the geographical location of Law Dome for readers who are not familiar with the region.

- Further information on handling/storage contamination was added in the revised manuscript in our response to reviewer 1. As requested, we added drilling information on p3, lines 8-9 "The sections of the core used in this work were extracted using thermal drilling (Morgan et al., 1997)."

- Reviewer wrote: "The relationship between Arctic and Antarctic pollutants concentration amount may be taken into consideration, but cannot become the key to any conclusions."

We agree with the reviewer, and it was our aim when writing this manuscript and its conclusions.

- Reviewer wrote "While it may be true that no earlier firn core studies are available to use as a guide for sample volume needed from Antarctica for OCPs, I think the volume used is really too big, a tenth could be enough for a more defined (in time) measurement."

Selection of our target volume was based on multiple factors including our method detection limits, previous concentrations found in other Antarctic matrices and volumes used in previous studies. Some previous Antarctic studies had used from 500 mL in surface snow (for HCB and HCHs, Kang et al. 2012) up to 132L of sea-ice (for a slightly wider range of OCPs, Dickhut et al. 2005). We could have used lower volumes to detect HCHs, but we were aiming at a wider range of OCPs.

The following has however been added in the revised manuscript: p7, lines 11-12 "HCHs and dieldrin were found at the largest concentrations suggesting that they could be targeted in future Antarctic glacial ice investigations to obtain more refined measurements using much lower volumes."

- C.E. is now defined.

Please find revised manuscript in attachment.

Please also note the supplement to this comment:
http://www.the-cryosphere-discuss.net/tc-2016-178/tc-2016-178-AC3-supplement.pdf

[Figure]

**Supplement:**

[revised manuscript text omitted]

---

## Author Response (AR1)

**Manuscript number**   tc-2016-178
**Manuscript type**     Brief Communication
**Title**               "Brief communication: Organochlorine pesticides in an
                        archived firn core from Law Dome, East Antarctica"

**Response to comments from Reviewer 1**

The authors would like to thank the anonymous reviewer for their very constructive comments.

*The paper describes the methods used and the results from analyzing two bulk samples from the upper sections from an ice core from Law Dome for Organochlorine pesticides (OCPs). There has been no similar studies from Antarctica published before so therefore this is an important contribution to our knowledge about the fate of contaminants in the atmosphere. The paper is well written and its main focus in on the analytical methods; basically how to handle ice core samples that has been archived for a long time. Naturally, this is important knowledge as such studies described here will likely be on similar situations; i.e. left-over material after some of the fundamental sampling and analyses for dating and climate studies have been completed.*

*However, before the paper is accepted I would like the authors to provide more and accurate glaciological information about the ice coring site which is completely lacking and are of importance for any studies using material from ice cores.*

*- The position of Law Dome is likely not familiar to many readers and it is important to include information about where it is situated, altitude, annual accumulation rate and mean annual temperature. All these factors are likely to have an impact of the fate of contaminants during transport, deposition and post-deposition and thus on the concentration found.*

> Site-specific information has been added from p2, line 26 to p3, line 2. "Law Dome is a small independent icecap located in Wilkes Land, East Antarctica and exposed to a maritime climate (Morgan et al., 1997). The DSS site is located near Law Dome summit, approximately 100 km from the coast and at 1370 m elevation (Morgan et al., 1997). This site was selected for its favourable bedrock topography and sufficiently low surface temperatures (mean annual average of - 21.8°C) which preclude summer melt (Morgan et al., 1997). The DSS site is characterised by a relatively high annual snow accumulation rate of 0.68 metres (water equivalent) (Roberts et al., 2015) facilitating the preservation of very clear seasonal cycles in glaciochemical species. This provided a means of accurate dating with monthly resolution in the upper portions of the core (van Ommen et al., 2004;Plummer et al., 2012;Roberts et al., 2015)."

*- From glaciological perspective Law Dome is not part of the "East Antarctic Ice Shelf"; an expression that is used in this paper which is not correct. There are many ice shelves around the Antarctic continent but they are separate from each other and thus do not form a unit, which is implied here. Also, despite its coastal location Law Dome is not part of any ice shelf. It is a local ice dome with an altitude of close to 1400 m which is much higher that any ice shelf and therefore it is very confusing when melting ice shelves are referred to both in the "Introduction" and "Conclusions". Furthermore, it is not unlikely that the sources and*

*transport paths to Law Dome and lower elevation ice shelves are different and thus it is not a good idea to mix these things freely in the discussion.*

> Thank you for the clarification. All references to the East Antarctic Ice Shelf have been removed from the manuscript.

*p.5, line 3-4. "Both samples, however, also contained traces of our storage and handling contamination markers, PBDEs, indicating that sampling and/or storage conditions of the archived firn cores introduced organic contamination." I would like to see some more information/ideas around this aspect. For instance, is the type of plastic bags used important?*

> We have further expanded on this point firstly on p5, lines 14-16 "These contaminants could have been introduced through various means such as the operator's personal clothing, storage in plastic bags, and the use of contaminated coring tools and inappropriate cleaning methods." Then on p7, line 28-29 "In particular, plastics should be avoided at all times as a general QA/QC measure for sampling and analysis of such compounds."

*Finally, "polar regions" should not be capitalized as now is done through the manuscript.*

> In the case of this manuscript, we purposely used capital letters for "Polar Regions" to distinguish "polar" from its other meanings in chemistry.

*I recommend that the paper is accepted after these comments are considered.*

**Response to comments from Reviewer 2**

The authors would like to thank the anonymous reviewer for their detailed comments and suggestions to improve this manuscript.

*This paper documents the results of analysis of organochlorine pesticides in a firn/ice core from Law Dome, Antarctica. The core was drilled several years before analysis, and was kept in storage. There are few studies documenting the accumulation of organic contaminants in ice (of any kind) from Antarctica that were transported by the atmosphere. The reasons for so few studies are many, including the lack of available cores, which itself is related to the difficulty with drilling, retrieving and storing cores from Antarctica. Another issue is the expense involved with analytical work. The work of the authors in this regard is greatly appreciated.*

*This paper should be given consideration for publication after addressing the general and specific comments made below.*

*General comments:*

*In my opinion, the paper addresses a relevant issue within the scope of TCD. The idea presented and tools used in the research are not novel, but the data are new and are a*

*contribution to understanding levels of contamination found in the Law Dome area of Antarctica.*

*The conclusions are generally sound, although rewriting should be considered. Yes, the results are from as early as 70 years ago, but that results from inherent poor dating resolution of organic contaminants in ice cores.*

> This part of the conclusion has been revised as "Results show that legacy OCPs could have accumulated at Law Dome in deep firn dated from as early as 70 years ago. Our current understanding of organic contaminant retention, mobility and fate during aging of snow and formation of ice is poor. Apparent concentrations in glacial ice may not accurately indicate historical deposition. The analysis of ice or firn cores may nevertheless give us valuable information on the current OCP reservoir in polar regions."

*The quantitative difference between OCP amounts found the Antarctic and the Arctic are important, but the authors need to be much more precise in P6L30+ where they state that "deposition rates are orders of magnitude lower than those from Arctic regions". In that statement, they are suggesting that the Arctic is one uniform region with regard to OCP deposition rates, and the published literature shows that this is not true (some of the publications showing this are not included in the reference list).*

> The reviewer is correct in that we had omitted to cite some key Arctic references (i.e. Hermanson et al. 2005 and Ruggirello et al. 2010) which are now included. These additional references were however considered in the initial version of this manuscript and in the statement that the reviewer has cited. We did not intend to suggest that the Arctic is one uniform region with regards to OCP deposition rates but simply wanted to point out the significantly lower concentrations and derived deposition rates that we are reporting in Antarctica compared to available Arctic literature.

> The sentence cited by the reviewer was modified to "OCP concentrations and deposition rates reported are orders of magnitude lower than those from sites with lower annual snow accumulation in the Arctic." This sentence was a conclusive statement based on previous discussion in the manuscript which was also extended to avoid possible confusion, see p5 lines 26-29 "Although the DSS site is characterised by higher snow deposition rates (0.68 m·yr$^{-1}$ weq) compared to the three other Arctic sites for which OCPs in glacial ice/firn cores were documented (from 0.36 to 0.52 m·yr$^{-1}$ weq), OCP results from the DSS site are 10 to 1000 fold lower than the reported Arctic levels (Hermanson et al., 2005;Isaksson et al., 2003;Ruggirello et al., 2010)."

*Within the conclusions, I disagree with the use of the word "in-situ" with respect to melting techniques being supported by the references used. I am not certain that the referenced investigations used "in-situ" melting the way it was done for the current investigation because not all of the references specifically state melting at a field site.*

> Our ice-melting unit was designed based on Gustafsson et al. 2005 who performed "in-situ" sampling of sea-ice from a ship, using a similar device that could not be used without ship support and crane lifting capabilities. Given that no other studies performed "in-situ" melting as we mean it, we removed all other references cited in the conclusions to avoid confusion and modified the paragraph accordingly.

*The scientific methods and assumptions are clearly stated, along with some of the limitations. I am a bit concerned that limitations involved with using the KISP have not been identified, including long running times and short battery life. However, these are only a problem when using the KISP in the field, instead of in a laboratory.*

> The KISP model that we used requires connection to mains electrical power, therefore it would require a generator if used directly on-site. Other KISP models have been fitted with batteries by the manufacturer. All KISPs are generally very energy efficient, so we do not expect this to be a major limitation. Presently, the best option would be to collect samples on site and bring them back to the closest operational research station as stated on p4 lines 10-11, although powering the system (both KISP and water bath) using generators could be an alternative provided resources are available. This is now mentioned on p8 line 1 "The melting process can be performed in-situ provided powering resources are available, or locally at the closest research station.".

> Increasing pumping rate is a possibility to reduce running times, however breakthrough of compounds would need to be investigated.

*The results have not been over-interpreted, which is important. The results here are very limited, but are still significant.*

*It would likely not be possible to reproduce these results, which is a common issue with environmental work on this level. There is very little related work in Antarctica, and it appears that the authors have given credit to previous work.*

*I have one question regarding the title: Is the DSS core really firn to a depth of 45.6 meters? There is no mention of the transition depth from firn to ice in this paper.*

> Yes, the section of the core that we used is firn as indicated by sample's densities between 0.63 and 0.74 $g \cdot cm^{-3}$ (P3L29).

*In the abstract, I do not understand what is meant by "nominal modern-use chemical contamination". How do deposition rates "orders of magnitude lower than those from Arctic regions" support validity? This is not standard QC procedure.*

> Our QC methods were limited by our inability to collect a "true field blank". We used modern-use chemicals (i.e. polybrominated diphenyl ethers, PBDEs) as an alternative to assess possible contamination of the core prior to the melting event. All information is documented in details in the method section of the manuscript. The abstract is only a brief summary of the content.

> The cited quote was modified to "nominal contamination by modern-use chemicals".

> The statement "orders of magnitude lower than those from Arctic regions" has been explained further in the body of the revised manuscript. See p5 lines 29-31 "This is consistent with the uneven distribution of the world's past usage (Voldner and Li 1995), with the Northern hemisphere having contributed larger emissions of OCPs than the Southern hemisphere, suggesting a larger pool of OCPs would have reached the Arctic in comparison to Antarctica." We believe it does not need expansion in the abstract due to word limits.

*In general, the paper is well organized and carefully written, with the issues noted elsewhere. I note that the use of C. E. in the paper (including the abstract) is not define, requiring that the reader understand what this means. That may not be the case.*

C.E. is now defined in the abstract and main manuscript.

*Figures: Figure 1 needs additional information, including the dimensions of the unit. It would be useful to know the grade of N2 used and its circulation within the device, and not just that it is a "clean atmosphere".*

$N_2$ grade has been added to the figure. This figure is a 2D conceptual drawing, therefore our options to add technical information such as dimensions are limited. In an effort to address the reviewer's comment, the unit dimensions and mass have been added to the method section of the manuscript, see p4 lines 3-4. This unit could be reproduced to any dimensions depending on intended use.

*There is a significant issue with references used in the paper. Why is the Legrand et al. 1984 reference used for aerosols in a glacier? Why not Murozomi et al., 1969 (GCA, 1969, 33, 1247-)? Murozomi et al. also had data for contaminant lead both from Greenland and Antarctica, and was the first paper to identify contamination of ice cores by long-range transport of an anthropogenic substance. This issue takes on greater relevance with the earlier studies about organic contaminants in the Arctic used as references in this manuscript. One assumes that the Gregor et al. (1995) paper is used as a reference because it was the earliest study on PCB deposition in an Arctic glacier. This assumption arises because of absence of later reference to PCBs in glaciers. So again, why use Legrand et al., 1984, as a reference if it is not the earliest?*

Many thanks for bringing this older reference to our attention. Legrand et al. 1984 has been replaced by Murozumi et al. 1969

*There is no reason (or reference) given for the quantification of PBDEs as evidence of contamination. What is the rationale behind this?*

The reason is given p4 lines 29 et seq. "True blanks (i.e. similar volume of archive samples representative of a deposition period pre-OCP production) were not available. In order to evaluate possible contamination, modern usage POPs, namely polybrominated diphenyl ethers (PBDEs), were used as markers of contamination resulting from the sampling, processing and/or storage of firn cores. PBDEs are commercial flame retarding compounds widely produced since the 1970's."

This is not a standard method and we only use PBDEs as "possible" indicators of contamination in an effort to address the absence of a true blank. We recognised that there is no direct link between PBDE contamination and OCP contamination in our discussion (see p5 line 21).

*Specific comments & technical issues:*

*P1L26: Replace "hereby" with "thereby". The statement made here, to "minimize the environmental and human health hazards that they pose" is overstated. Even though the*

*compounds on the Stockholm list have been banned or restricted, they are still found in the environment. And they are still moving around. The only thing Stockholm can accomplish is prevent the mass of these contaminants now in the environment from becoming greater.*

*P1L33: Again, Legrand et al., 1984 is not the best reference in this context.*

*P2L5: The claim that "only one study has documented OCP concentrations in glacial ice/firn from the Arctic" is not correct. I can immediately think of 3 without looking.*

*P3L4: This repeats P2L2.*

*P3L19: C. E.?*

*P4L27: The original results were not "corrected to estimate the mean deposition". It is not a matter of correction, but calculation.*

All above specific comments were addressed as suggested.

*P2L6: To say that there are no OCP studies in firn/ice cores from Antarctica is splitting hairs a bit too much for work like this. What about Kang et al., 2012, from the reference list?*
*P5L9: Again, this limited selection of references is a bit surprising considering those not mentioned.  Why are no comparisons offered between the results of Kang et al. 2012 and the results of this manuscript?*

We referenced Kang et al. 2012 in other places in this manuscript. In the specific statement (p2 line 5) we are referring to "firn/ice cores". Kang et al. studied surface snow, therefore their study is not directly comparable. We however added a comparison of their HCH results on p6 lines19-21 "Concentrations reported in this study are 19 and 22 pg·L$^{-1}$ for α-HCH  and 22 and 60 pg·L$^{-1}$ for γ-HCH, which are in the lower range of findings in surface snow collected more recently in Antarctica (Kang et al., 2012)."

*P3L13: Apparently it is true that Isaksson et al. (2003) never mention the diameter of the core used.  The current manuscript also never mentions the diameter of the Law Dome core.*

In this particular sentence, we are looking for volumes of ice analysed in the literature. Isaksson et al. 2003 does not indicate the sample volumes, only the length of the core, which is not sufficient to infer a volume.

Nevertheless, we have now indicated the diameter of the DSS core (p3 line 9).

*P3L15: While it may be true that no earlier firn core studies are available to use as a guide for sample volume needed from Antarctica, the authors could have used Kang et al. study on surface snow as a guide.*

We considered Kang et al., as well as other Antarctic studies on other matrices. They are all listed in this same paragraph (p3 line 20).

*P3L26: Was the system shown in Figure 1 capable of holding 144 L of melt?*

The system would have been capable of holding 144L of liquid water but was not capable of holding the corresponding frozen volume. We melted cores in two successive batches for each sample. This was added p4 lines 4-5 "These were

obtained through melting of two successive batches for each due to the limited capacity of the melting unit."

*P4L28: In my dictionary, "basal area" is defined as the area of total tree trunks (diamters) as a fraction of given land area where the trees are growing. That does not seem to apply here.*

We replaced "basal area" by "surface area".

*P4L31: What is meant by "dissolved fraction of the melt water"?*

The paragraph on the filtering materials p4 lines 14-17 was modified to describe and delineate dissolved and particle fractions.

*P6L7: Reference to Stockholm for ban on Dieldrin is not very good. Dieldrin was banned under other regulations many years before 2004.*

The Stockholm Convention is the official international treaty that banned dieldrin globally. Little detailed information is available about dieldrin restrictions for individual nations. Please note that in the same sentence we mention that its usage was restricted from the 1970s.

*P7L8: The authors need to do a better job describing "in situ".*

The paragraph was revised in an attempt to address this comment, see p7 lines 29 et seq. "The ice-melting unit used for the present work is designed to be transportable and represents a comprehensive tool offering a means of storage, transport, melting and pumping of large volumes of ice. It provides a cost-effective solution to the logistical challenges of transporting these volumes of ice back from Antarctica. It also reduces risks for contamination during storage and transport as well as minimising overall handling of the samples. The melting process can be performed in-situ provided appropriate power sources are available, or locally at the closest research station."

**Response to comments from Reviewer 3**

The authors would like to thank the anonymous reviewer for their thoughtful comments and suggestions.

*In order to understand better the magnitude and to limit the environment's damage, it is clearly necessary to know about transportation, accumulation and concentration changes of organic pollutants through ages. One irreplaceable source to obtain some of this information is the investigation of chemical compounds in polar ice cores. Since snowflakes have the capacity to adsorb gas-phase chemical compounds, ice-sheets represent an archive of notions of the varying deposition of trace chemicals. From the stratigraphy of the ice cores, the ice-sheets can be temporally related, and every core can cover the chemical history of many decades. Antarctica's low temperatures during the whole year both reduce the speed of chemical reactions and allow the ice layering mentioned above. The paper, well written, describes the methods and the results from two bulk samples from the upper sections from an*

*"old" ice core from Law Dome for organochlorine pesticides: few studies have been published, so the paper is another small step towards knowledge about contaminants transport mechanism in the atmosphere.*

*The ice core used in this study was drilled on Law Dome, a small ice cap with independent ice flow located on the edge of the main East Antarctic ice sheet (a map could be useful for many readers). The characteristics of Dome Summit South (DSS), include a high annual accumulation rate (0.7 m/yr ice equivalent), relatively low mean surface temperatures (-21.8 ∘C), and low wind speeds (8.3 m/s). These site characteristics lead to highly resolved records with clear annual cycles in most measured parameters, giving very robust chronological control. I did not find any of this information on the paper, and I have to find myself, in the literature, despite their fundamental importance.*

> Thank you for your suggestions. Information on site-specific characteristics has been added to the revised manuscript as previously addressed in our response to Reviewer 1.

> We have added a map (see new Figure 1) which gives a better idea of the geographical location of Law Dome for readers who are not familiar with the region.

*As RC1 wrote, it is also absolutely important to have more information around storage/handling contamination, but not only, the sampling conditions must absolutely be considered and described: was it a manual drilling or not? I think not, the ice-drilling was too deep (1196 m), this "modus operandi" can have influenced the results.*

> Further information on handling/storage contamination was added in the revised manuscript in our response to reviewer 1.
> As requested, we added drilling information on p3, lines 8-9 "The sections of the core used in this work were extracted using thermal drilling (Morgan et al., 1997)."

*The relationship between Arctic and Antarctic pollutants concentration amount may be taken into consideration, but cannot become the key to any conclusions.*

> We agree with the reviewer, and it was our aim when writing this manuscript and its conclusions.

*While it may be true that no earlier firn core studies are available to use as a guide for sample volume needed from Antarctica for OCPs, I think the volume used is really too big, a tenth could be enough for a more defined (in time) measurement.*

> Selection of our target volume was based on multiple factors including our method detection limits, previous concentrations found in other Antarctic matrices and volumes used in previous studies. Some previous Antarctic studies had used from 500 mL in surface snow (for HCB and HCHs, Kang et al. 2012) up to 132L of sea-ice (for a slightly wider range of OCPs, Dickhut et al. 2005). We could have used lower volumes to detect the HCHs, but we were aiming at a wider range of OCPs.

> The following has however been added in the revised manuscript: p7, lines 11-12 "HCHs and dieldrin were found at the largest concentrations suggesting that they

could be targeted in future Antarctic glacial ice investigations to obtain more refined measurements using much lower volumes.."

*P3L19: C. E.?*

C.E. is now defined.

*I recommend that the paper is accepted after these comments are considered.*

.

[revised manuscript text omitted]